# The temporospatial epidemiology of rheumatic heart disease in Far North Queensland, tropical Australia 1997–2017; impact of socioeconomic status on disease burden, severity and access to care

**Katherine Kang**[1], **Ken W.T. Chau**[1], **Erin Howell**[2], **Mellise Anderson**[2], **Simon Smith**[1], **Tania J. Davis**[3], **Greg Starmer**[3], **Josh Hanson**[1,4]*

**1** Department of Medicine, Cairns Hospital, Cairns, Queensland, Australia, **2** Rheumatic Heart Disease Program, Tropical Public Health Unit, Cairns, Queensland, Australia, **3** Department of Cardiology, Cairns Hospital, Cairns, Queensland, Australia, **4** Kirby Institute, University of New South Wales, Sydney, New South Wales, Australia

* jhanson@kirby.unsw.edu.au

**Data Availability Statement:** Data cannot be shared publicly because of the Queensland Public

## Abstract

### Background

The incidence of rheumatic heart disease (RHD) among Indigenous Australians remains one of the highest in the world. Many studies have highlighted the relationship between the social determinants of health and RHD, but few have used registry data to link socioeconomic disadvantage to the delivery of patient care and long-term outcomes.

### Methods

A retrospective study of individuals living with RHD in Far North Queensland (FNQ), Australia between 1997 and 2017. Patients were identified using the Queensland state RHD register. The Socio-Economic Indexes for Areas (SEIFA) Score–a measure of socioeconomic disadvantage–was correlated with RHD prevalence, disease severity and measures of RHD care.

### Results

Of the 686 individuals, 622 (90.7%) were Indigenous Australians. RHD incidence increased in the region from 4.7/100,000/year in 1997 to 49.4/100,000/year in 2017 (p<0.001). In 2017, the prevalence of RHD was 12/1000 in the Indigenous population and 2/1000 in the non-Indigenous population (p<0.001). There was an inverse correlation between an area's SEIFA score and its RHD prevalence (rho = -0.77, p = 0.005).

249 (36.2%) individuals in the cohort had 593 RHD-related hospitalisations; the number of RHD-related hospitalisations increased during the study period (p<0.001). In 2017, 293 (42.7%) patients met criteria for secondary prophylaxis, but only 73 (24.9%) had good

Health Act 2005. Data are available from the Far
North Queensland Human Research Ethics
Committee (contact via email
Cairns_Ethics@health.qld.gov.au) for researchers
who meet the criteria for access to confidential
data. This is because the data are obtained from
the Queensland State RHD register and these data
are not available publicly.

**Funding:** The authors received no specific funding
for this work.

**Competing interests:** The authors have declared
that no competing interests exist.

adherence. Overall, 119/686 (17.3%) required valve surgery; the number of individuals having surgery increased over the study period (p = 0.02).

During the study 39/686 (5.7%) died. Non-Indigenous patients were more likely to die than Indigenous patients (9/64 (14%) versus 30/622 (5%), p = 0.002), but Indigenous patients died at a younger age (median (IQR): 52 (35–67) versus 73 (62–77) p = 0.013). RHD-related deaths occurred at a younger age in Indigenous individuals than non-Indigenous individuals (median (IQR) age: 29 (12–58) versus 77 (64–78), p = 0.007).

## Conclusions

The incidence of RHD, RHD-related hospitalisations and RHD-related surgery continues to rise in FNQ. Whilst this is partly explained by increased disease recognition and improved delivery of care, the burden of RHD remains unacceptably high and is disproportionately borne by the socioeconomically disadvantaged Indigenous population.

### Author summary

Rheumatic heart disease (RHD), a disease of poverty and disadvantage, is almost completely preventable. It is now extremely rare in wealthy countries, but in Far North Queensland in tropical Australia, the incidence of RHD, RHD-related hospitalisations and RHD-related surgery is continuing to rise, with the burden of disease borne almost entirely by the region's Indigenous population. While the increasing incidence of RHD and its complications may be partly explained by improvements in local service delivery, the disease remains inextricably linked to socioeconomic disadvantage. In this study, not only were patients living in socioeconomically disadvantaged areas more likely to have RHD, but they were also paradoxically less likely to receive valve surgery. The current local model of care—which is centralised, medical and emphasises disease monitoring and secondary prophylaxis—appears to be having a limited impact on morbidity. Strategies must evolve—in partnership with Indigenous communities—to have a greater focus on disease prevention by addressing the personal, community and environmental factors that increase the risk of the disease. This is likely to not only reduce the incidence of RHD, but will also tend to reduce the burden of the many other diseases that result from socioeconomic disadvantage and that disproportionately affect Indigenous Australians.

## Introduction

Rheumatic heart disease (RHD)—a chronic, debilitating and potentially fatal disease of social disadvantage—is almost entirely preventable [1–3]. Improvements in the standard of living and the development of more effective treatment of group A streptococcal (GAS) infections during the 20th century, resulted in the near elimination of acute rheumatic fever (ARF), which in turn, led to a marked reduction in the burden of RHD in high-income countries [4,5]. A similar trend has been observed in the non-Indigenous Australian population. However, the prevalence of RHD in some Indigenous Australian communities remains amongst the highest in the world, a source of continuing national shame [6,7].

"Indigenous Australians" is a frequently used term, but it does not account for the enormous variation in circumstances that exists for Indigenous individuals in different parts of the

country [8]. Although socioeconomic disadvantage is more common among Indigenous Australians generally, there are distinct challenges faced by Indigenous people who live in rural and remote locations, where access to health care is limited and where other health conditions compete for finite resources [8–10]. It is also important to note that while Torres Strait Islander Australians are Indigenous Australians, they are ethnologically distinct to Aboriginal Australians and have their own unique cultural history. Although there is some overlap between the health challenges faced by the two peoples, there are also many differences [8].

RHD is estimated to kill over 300,000 people globally every year, but the disease remains profoundly neglected [11,12]. Although medical interventions can reduce the morbidity and mortality of established RHD, interventions that reduce the incidence of the inciting GAS infections–primordial prevention—are likely to be more cost-effective, and there is a growing recognition of the role of the socioeconomic factors in the development of RHD, even among clinicians [7]. However, while many studies have highlighted the relationship between the social determinants of health and the burden of RHD, most have been observational, cross-sectional and have not linked socioeconomic status to interventions or long-term outcomes [3]. This can be at least partly explained by the fact that RHD usually occurs in resource-poor settings where registry data—which provide more detail on delivery of care and longitudinal follow-up—are frequently lacking [11].

ARF has been a notifiable disease in the state of Queensland since 1999, while RHD has been notifiable since 2018. Affected patients' demographic and clinical data are collected by staff employed by a dedicated RHD program, stored in a database—the RHD register—which is used to assist the coordination of medical care. Far North Queensland (FNQ), in tropical Australia is home to approximately 280,000 people who are dispersed across an area of 380,748 km$^2$. No fewer than 17% of the local population identify as Indigenous Australians, almost half of whom report Torres Strait Islander heritage [13]. Evolving prosperity and public health interventions have seen the elimination of malaria and filariasis in the region, while the incidence of several others–including hepatitis B, strongyloidiasis and leprosy–is in steep decline [14–16]. However it is also a region which still contains 3 of the 10 most socio-economically disadvantaged local government areas in Australia, all 3 are communities with a predominantly Indigenous population [17]. When compared to other regions of Australia, there are relatively few studies that have examined the local prevalence of RHD and almost none that compare the relative burdens in Aboriginal and Torres Strait Islander Australians [18,19].

This study therefore used registry data to examine the temporospatial epidemiology of RHD in FNQ and the performance of the local RHD control program in addressing the disease. There was a focus on the relationship between measures of socioeconomic disadvantage and disease prevalence, severity, and treatment. The study sought to examine differences in disease burden between Aboriginal and Torres Strait Islander Australians and to evaluate the care of individuals living in rural and remote—rather than urban—locations. It was hoped that these data might be used to inform local strategies to address the disease in a more comprehensive manner.

## Methods

### Ethics statement

The Far North Queensland Human Research Ethics Committee provided ethical approval for the study (HREC/18/QCH/91–1261). As the data were retrospective and de-identified, the Committee waived the requirement for informed consent.

## Study design

This retrospective study was performed at Cairns Hospital, the sole tertiary referral hospital in FNQ. There was also significant contribution from the local Rheumatic Heart Disease Program which aims to improve the local care of ARF and RHD. The Queensland RHD register was used to identify patients, who were eligible for inclusion in the study if they had a diagnosis of RHD confirmed on echocardiogram that had been reported by a specialist physician, between January 1, 1997 and December 31, 2017. RHD was further categorised—based on the specialist physicians' echocardiogram report—as mild, moderate or severe. Patients who were reported as having "borderline" disease (minor abnormalities identified by echocardiography that could represent normal variation) were not included in the study. Patients were said to having a history of ARF if they had had an episode of possible, probable or confirmed ARF reported to the RHD register; ARF was diagnosed by local clinicians using their clinical judgment, supported by the Jones criteria which were updated during the study period [7].

The patients' demographics and their clinical course during the study period were collected from the database and their medical records. The patients' Indigenous status was recorded; when individuals register with the public health system, they are routinely asked whether they identify as an Aboriginal Australian, a Torres Strait Islander Australian, both or neither. Australian Bureau of Statistics population data were used to calculate disease incidence and prevalence [13]. If an individual lived in the region's administrative hub—Cairns—they were said to have an urban address, otherwise they were deemed to live in a rural or remote area. Socioeconomic status was quantified using the Socio-Economic Indexes for Areas (SEIFA) Score, a measure of socioeconomic disadvantage developed by the Australian Bureau of Statistics [20].

Complete secondary prophylaxis data were only available from 2007 and so adherence to secondary prophylaxis could only be assessed after this date. Adherence to parenteral penicillin was determined by dividing the number of doses of penicillin received by the number prescribed; ≥10 parenteral penicillin doses in 1 year was defined as good adherence. Hospitalisation data were recorded with the presentations classified using International Statistical Classification of Diseases and Related Health Problems (ICD-10) coding. Hospitalisations were defined as being RHD related if it could be explained by valvular pathology or its complications. All cardiac surgical interventions (valvuloplasty and valve replacement) were recorded. If a patient died, their cause of death was identified by review of the medical record where this was accessible.

## Statistics

Data were collected from the RHD register or the medical record, de-identified, entered into an electronic database (Microsoft Excel 2016, Microsoft, Redmond, WA, USA) and analysed using statistical software (Stata version 14.2, StataCorp LLC, College Station, TX, USA). Groups were analysed using the Kruskal-Wallis and chi-squared tests, where appropriate. Correlation coefficients were determined using Spearman's method. Trends over time were determined using an extension of the Wilcoxon rank-sum test [21]. Multivariate analysis was performed using backwards stepwise logistic regression. If individuals were missing data, they were not included in analyses which evaluated those variables.

## Results

### Demographics

There were 686 individuals diagnosed as living with RHD in the region during the study period. Their median (interquartile range (IQR)) age at the time of RHD diagnosis was 29

(17–44) years, 458 (66.7%) were female and 622 (90.7%) identified as Indigenous Australians. Among the 622 Indigenous patients, 347 (55.8%) identified as Aboriginal Australians, 205 (32.9%) identified as Torres Strait Islander Australians while 70 (11.3%) identified as both. Among the 64 non-Indigenous patients, 29 (45.3%) were born in Australia, while 25 (39%) were born in other countries from the Asia-Pacific region.

Indigenous patients were younger than non-Indigenous patients (median (IQR): 33 (23–47) versus 60 (27–74), p<0.001), more likely to live in a rural or remote location (333/622 (53.5%) versus 11/64 (17.2%), p<0.001) and more likely to live in a socioeconomically disadvantaged area (median (IQR) SEIFA: 870 (836–967) versus 967 (925–967), p<0.001.

### Incidence of acute rheumatic fever

There was a documented history of ARF in 358/686 (52.2%); there was no reduction in new ARF diagnoses during the study period (p for trend = 0.46) (Fig 1).

### Incidence of RHD

The incidence of new RHD diagnoses increased in the region from 4.7/100,000/year in 1997 to 49.4/100000/year in 2017 (p for trend<0.001). The incidence of RHD was higher in the Indigenous population than the non-Indigenous population throughout the study period (Fig 2). In 2017, the incidence of new RHD diagnoses was 255/100,000/year in Indigenous population compared with 3/100,000/year in the non-Indigenous population (p<0.0001).

RHD incidence increased in both urban and rural and remote locations, but it was higher in rural and remote locations throughout the study (Fig 3). In 2017 the incidence was 298/

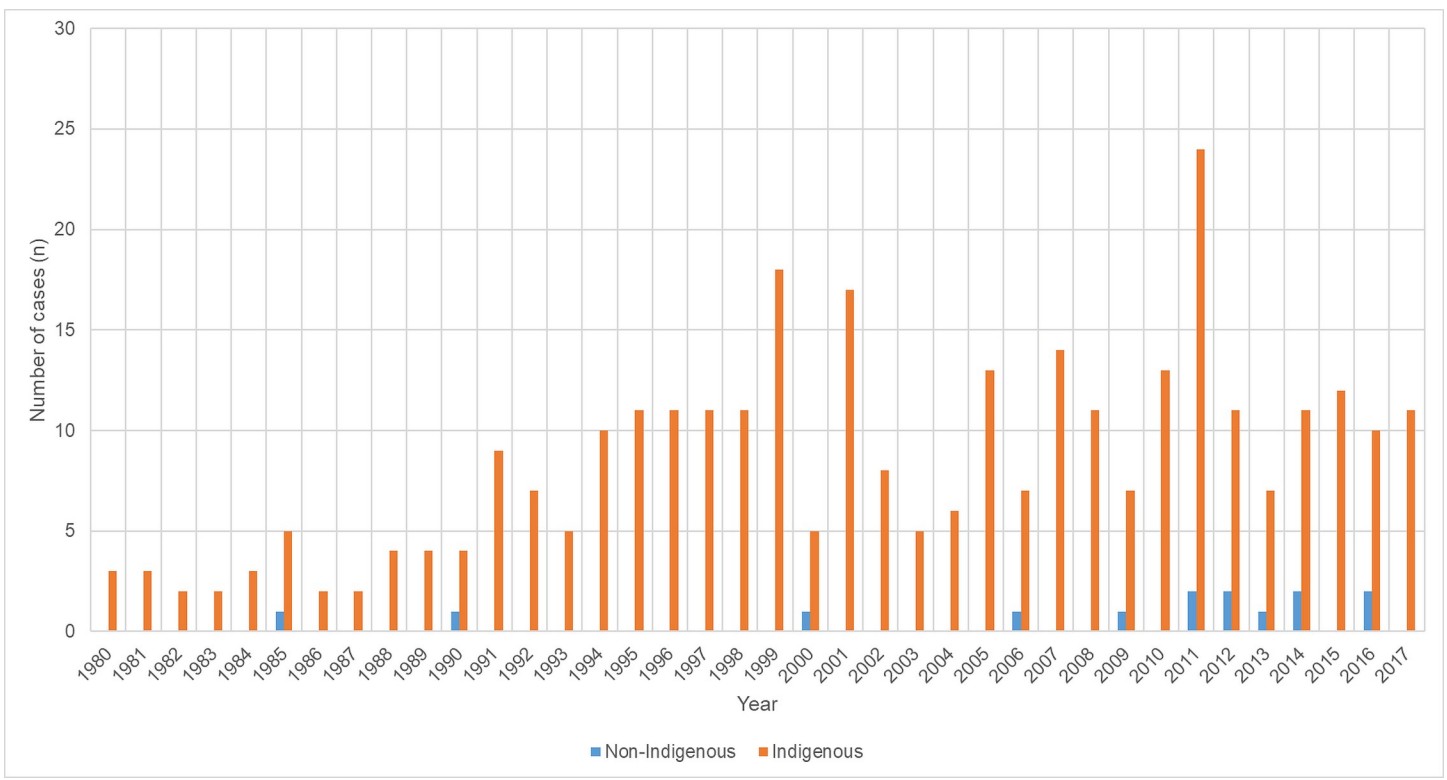

**Fig 1. Number of new cases of ARF diagnosed in the cohort annually after 1980, stratified by Indigenous status.**

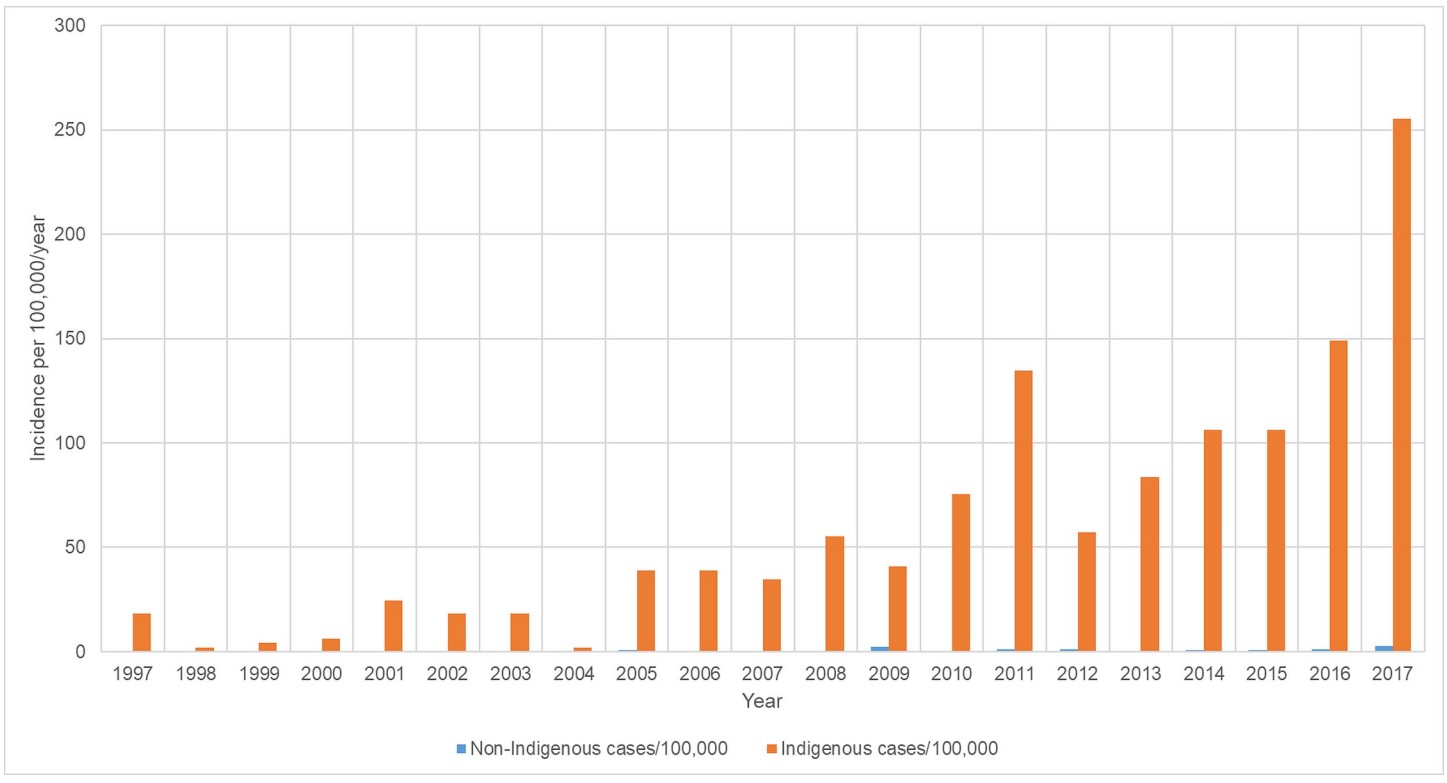

**Fig 2. Incidence of new RHD diagnosis during the study period, stratified by Indigenous status**

100,000/year in rural and remote locations compared with 23/100,000/year in the urban population (p<0.001).

## Prevalence

At the end of the study period, the prevalence of RHD in the Indigenous population was 12/1000 compared with 2/1000 in the non-Indigenous population (p<0.001). The prevalence was also higher in individuals living in rural and remote regions in those with urban address (12/1000 versus 1/1000 p<0.001).

## Spatial epidemiology of disease prevalence and association with socioeconomic status

There was significant heterogeneity in the prevalence of RHD within the region (Fig 4). The prevalence varied from 1/1000 in the southern region to 27/1000 population in Western Cape York. There was an inverse correlation between an area's SEIFA score and its RHD prevalence (Spearman's rho = -0.77, p = 0.005) (Fig 5).

## Disease severity

Indigenous patients, patients living in rural and remote locations and patients living in areas of greater socioeconomic disadvantage were less likely to have severe disease (Tables 1, 2 and 3). At the end of the study, 313/647 (48.4%) living patients had mild RHD, 176 (27.2%) had moderate and 158 (24.4%) had severe disease. Among the 592 living Indigenous patients, 138 (23.3%) had severe disease compared with 20/55 (36.4%) non-Indigenous patients (p = 0.03),

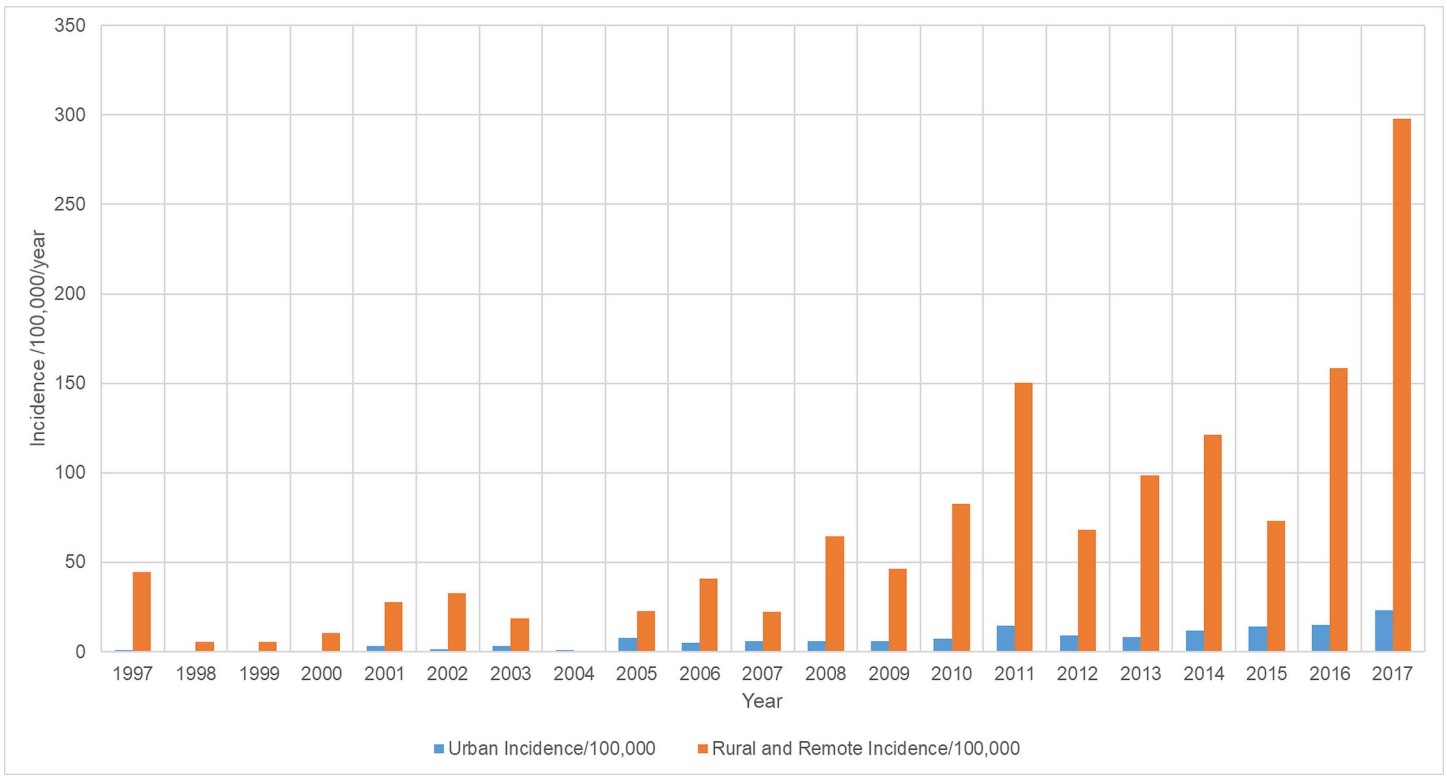

**Fig 3. Incidence of new RHD diagnosis during the study period, stratified by residential address**

however, Indigenous patients with severe disease were younger than non-Indigenous patients with severe disease (median (IQR): 41 (29–54) versus 62 (34–71) years, p = 0.005). In multivariate analysis that considered age, Indigenous status, SEIFA score and residence in a rural/remote location, only age (odds ratio (OR): 1.03 (95% Confidence interval (CI):1.02–1.04, p<0.001) and rural/remote residence were associated with disease severity (OR (95%CI): 0.63 (0.43–0.91), p = 0.01) at the end of the study period.

### Service delivery and retention in care

Patients living in rural and remote locations and patients living in areas of greater socioeconomic disadvantage were more likely to receive specialist review (Tables 2 and 3). A greater proportion of Indigenous patients received specialist review than non-Indigenous patients, however the difference failed to reach statistical significance (Table 1)

Patients had a median (IQR) of 5 (3–8) echocardiograms over median (IQR) follow-up time of 9 (5–16) years. Among the 629 (91.6%) patients who had more than one echocardiogram performed during the study period, RHD progressed in 131 (20.8%), was stable in 443 (79.4%) and improved in 55 (8.7%) over a median (IQR) of 8 (4–14) years. The risk of disease progression was not influenced by Indigenous status, rural/remote residence, or SEIFA score (Tables 1, 2 and 3). Only 189/686 (27.6%) had documented dental review in the public health system.

### Secondary prophylaxis

From 2007—the year when data were available—a total of 388 (57%) patients were prescribed secondary prophylaxis, 367 (95%) received parenteral penicillin while 21 (5%) received oral

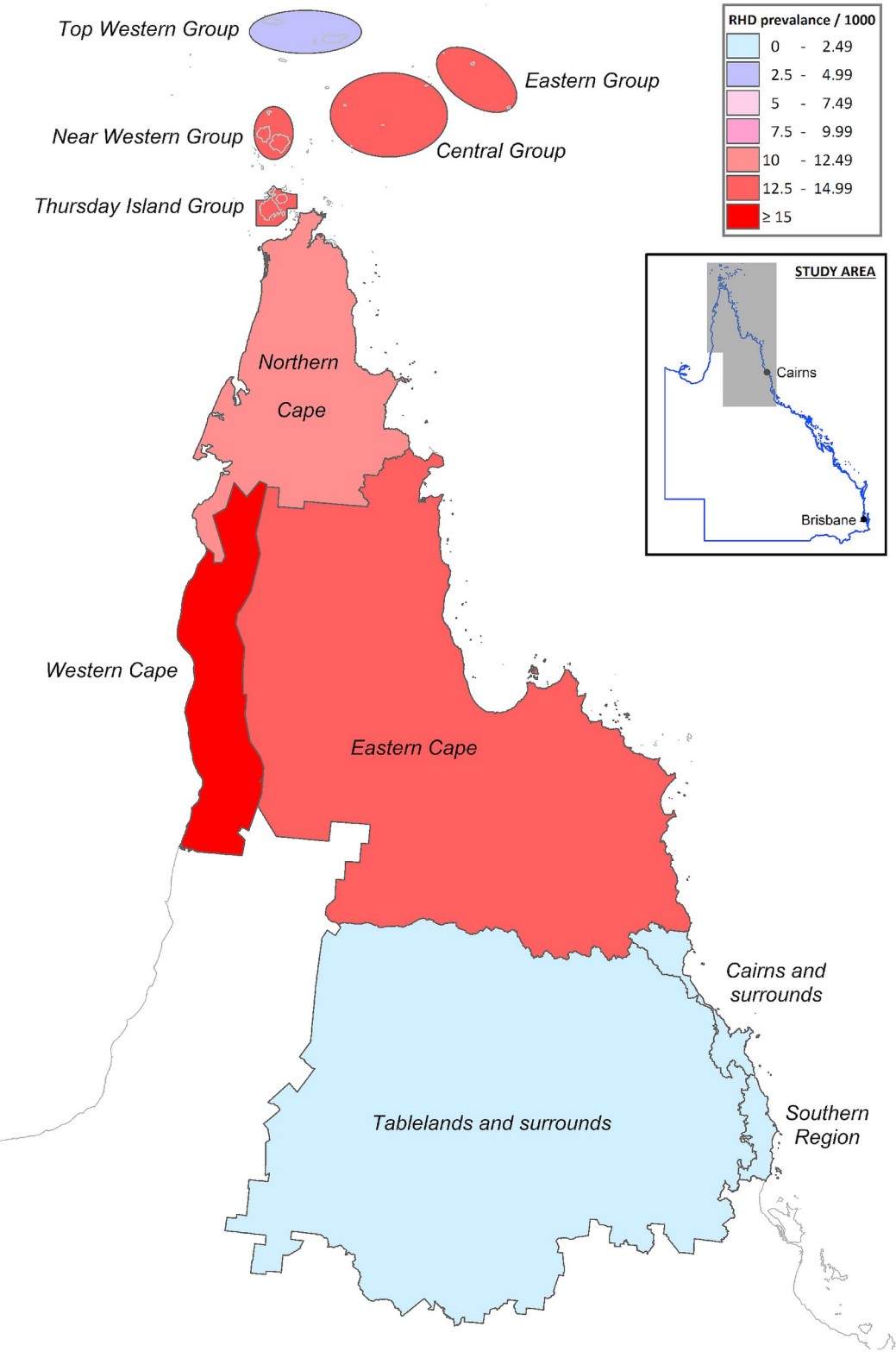

**Fig 4. Regional prevalence of RHD in Far North Queensland.** The map was created using constructed using mapping software (MapInfo version 15.02, Connecticut, USA) using data provided by the State of Queensland (QSpatial). Queensland Place Names—State of Queensland (Department of Natural Resources, Mines and Energy) 2019, available under Creative Commons Attribution 4.0 International licence https://creativecommons.org/licenses/by/4.0/. 'Coastline and state border–

Queensland—State of Queensland (Department of Natural Resources, Mines and Energy) 2019, available under Creative Commons Attribution 4.0 International licence https://creativecommons.org/licenses/by/4.0/.

therapy. During this time, the median (IQR) adherence for parenteral prophylaxis was 49% (34–63). There was no difference in adherence between Indigenous and non-Indigenous patients (median (IQR): 41 (23–58) versus 46 (10–60), p = 0.90). However, patients living in a rural or remote location had greater adherence than urban patients (median (IQR): 48% (25–62) versus 38% (19–53), p = 0.0001) and SEIFA score was inversely associated with adherence (Spearman's rho = -0.13, p = 0.002). Patients with disease progression on sequential echocardiograms were likely to have lower adherence to secondary prophylaxis (median (IQR): 35% (19–58) versus 44% (26–60), p = 0.03). At the end of the study period, 293/686 (42.7%) met criteria for secondary prophylaxis, 79 (24.9%) of whom had good adherence.

## Hospitalisations

During the study 627/686 (91.4%) were hospitalised, in 249 (39.7%) at least one of the hospitalisations was RHD-related. There were 593 RHD-related hospitalisations among these 249 patients; the number of RHD-related hospitalisations increased over the study period (p<0.001) (Fig 6). 218/249 (87.6%) hospitalisations were related to cardiac manifestations alone, 12/249 (4.8%) had only neurological hospitalisations, while 19/249 had both cardiac and neurological hospitalisations. There were 50 strokes (38 (76%) ischaemic and 12 (24%) haemorrhagic) among the 31 patients with a neurological admission.

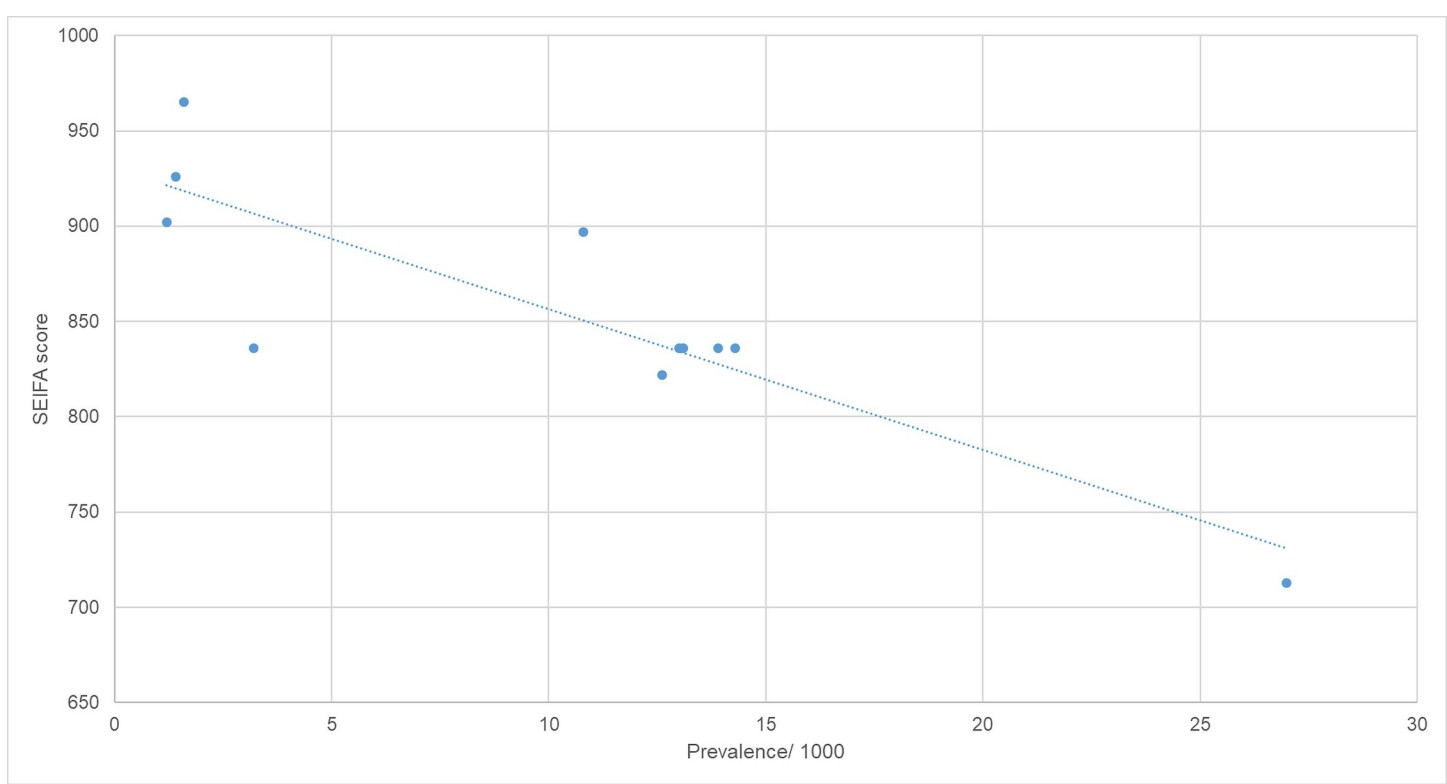

**Fig 5. Prevalence of RHD by socioeconomic status (determined using SEIFA score [20]).**

**Table 1. Characteristics of the cohort stratified by Indigenous status.**

| Variable | Indigenous n = 622 | Non-Indigenous n = 64 | p |
|---|---|---|---|
| Age at the end of the study | 33 (23–47) | 60 (27–74) | 0.0001 |
| Age at RHD diagnosis | 28 (17–43) | 54 (24–68) | 0.0001 |
| Female | 420 (67.5%) | 38 (59.4%) | 0.18 |
| Rural/remote residence | 333 (53.5%) | 11 (17.2%) | <0.0001 |
| Severe disease | 151 (24.3%) | 26 (40.6%) | 0.004 |
| Hospitalisations/person | 5 (2–10) | 6 (2–16) | 0.73 |
| RHD-related hospitalisations | 215 (35%) | 34 (62.5%) | 0.003 |
| Valve surgery or valvuloplasty | 98 (15.8%) | 21 (32.8%) | 0.001 |
| Died | 30 (4.8%) | 9 (14.1%) | 0.002 |
| Age at death | 54 (34–68) | 74 (62–77) | 0.01 |
| Adherence | 41 (23–58) | 46 (10–60) | 0.54 |
| Good adherence | 67/274 (24.5%) | 6/19 (31.6%) | 0.58 |
| Number of echocardiograms | 5 (3–8) | 4 (2–8) | 0.18 |
| RHD progression on serial echocardiograms | 121/575 (21%) | 10/54 (18.5%) | 0.66 |
| Specialist review | 585 (94.1%) | 58 (90.6%) | 0.28 |
| Dental review | 186 (29.9%) | 3 (4.7%) | <0.0001 |

All numbers are median (interquartile range) or the absolute number (%)

**Surgical management.** During the study period 119 (17.4%) individuals had 150 episodes of surgery during which 215 individual procedures were performed; 152/215 (70.7%) procedures were valve replacements and 63 (29.3%) were valvuloplasties (Table 4). The number of individuals having surgery increased during the study period (p for trend = 0.02). (Fig 7)

Surgical procedures were performed in 98/622 (15.8%) Indigenous patients and 21/64 (32.8%) non-Indigenous patients (p = 0.001) and in 42/344 (12.2%) of patients living in remote

**Table 2. Characteristics of the cohort stratified by residence in a rural/remote or an urban location.**

| Variable | Rural/Remote n = 344 | Urban n = 342 | p |
|---|---|---|---|
| Age at the end of the study | 35 (24–50) | 34 (22–50) | 0.23 |
| Age at RHD diagnosis | 29 (18–45) | 28 (16–44) | 0.16 |
| Female | 224 (65.1%) | 234 (68.4%) | 0.36 |
| Indigenous Australian | 333 (96.8%) | 289 (84.5%) | <0.0001 |
| Severe disease | 76 (22.1%) | 101 (29.5%) | 0.03 |
| Hospitalisations/person | 5 (2–9) | 5 (3–12) | 0.11 |
| RHD-related hospitalisations | 104 (30.2%) | 145 (42.4%) | 0.001 |
| Valve surgery or valvuloplasty | 42 (12.2%) | 77 (22.5%) | <0.0001 |
| Died | 17 (4.9%) | 22 (6.4%) | 0.40 |
| Age at death | 55 (30–70) | 61 (43–71) | 0.60 |
| Adherence | 45 (28–62) | 38 (19–53) | 0.02 |
| Good adherence | 38/139 (27%) | 35/154 (22.7%) | 0.36 |
| Number of echocardiograms | 5 (3–9) | 5 (3–8) | 0.32 |
| RHD progression on serial echocardiograms | 66/319 (20.7%) | 65/310 (21.0%) | 0.93 |
| Specialist review | 333 (96.8%) | 310 (90.6%) | 0.001 |
| Dental review | 134 (39.0%) | 55/342 (16.1%) | <0.0001 |

All numbers are median (interquartile range) or the absolute number (%)

**Table 3. Association of socioeconomic status and demographic characteristics, and the features, management, and clinical course of the RHD.**

| Variable | Yes (n) | No (n) | SEIFA if yes Median (IQR) | SEIFA if no Median (IQR) | p |
|---|---|---|---|---|---|
| Age <40 years | 410 | 276 | 870 (836–967) | 870 (771–967) | 0.26 |
| Female | 458 | 228 | 870 (836–967) | 870 (836–967) | 0.21 |
| Indigenous Australian | 622 | 64 | 870 (836–967) | 967 (924–967) | 0.0001 |
| Remote/rural residence | 344 | 342 | 836 (713–853) | 967 (914–967) | 0.0001 |
| Severe disease | 177 | 509 | 914 (836–967) | 870 (836–967) | 0.02 |
| >10 hospitalisations | 190 | 496 | 922 (836–967) | 870 (836–967) | 0.004 |
| RHD related hospitalisation | 249 | 437 | 914 (836–967) | 859 (771–967) | 0.0001 |
| Surgical intervention | 119 | 567 | 958 (859–967) | 870 (836–967) | 0.0001 |
| Died | 39 | 647 | 931 (836–967) | 870 (836–967) | 0.17 |
| Good Adherence to secondary prophylaxis | 73 | 220 | 870 (836–967) | 896 (836–967) | 0.85 |
| ≥ 5 echocardiograms | 404 | 282 | 870 (836–967) | 914 (836–967) | 0.14 |
| RHD progression on serial echocardiograms | 131 | 498 | 870 (771–967) | 870 (836–967) | 0.54 |
| Specialist review | 643 | 43 | 870 (836–967) | 967 (836–967) | 0.0003 |
| Dental review | 189 | 497 | 859 (771–958) | 896 (836–967) | 0.0001 |

IQR: interquartile range; RHD: Rheumatic heart disease

locations compared with 77/342 (22.5%) with an urban address (p<0.001). The median (IQR) SEIFA score was higher in patients receiving surgery (958 (859–967)) than those that did not (870 (836–967)), p<0.001. In multivariate analysis that considered disease severity, Indigenous

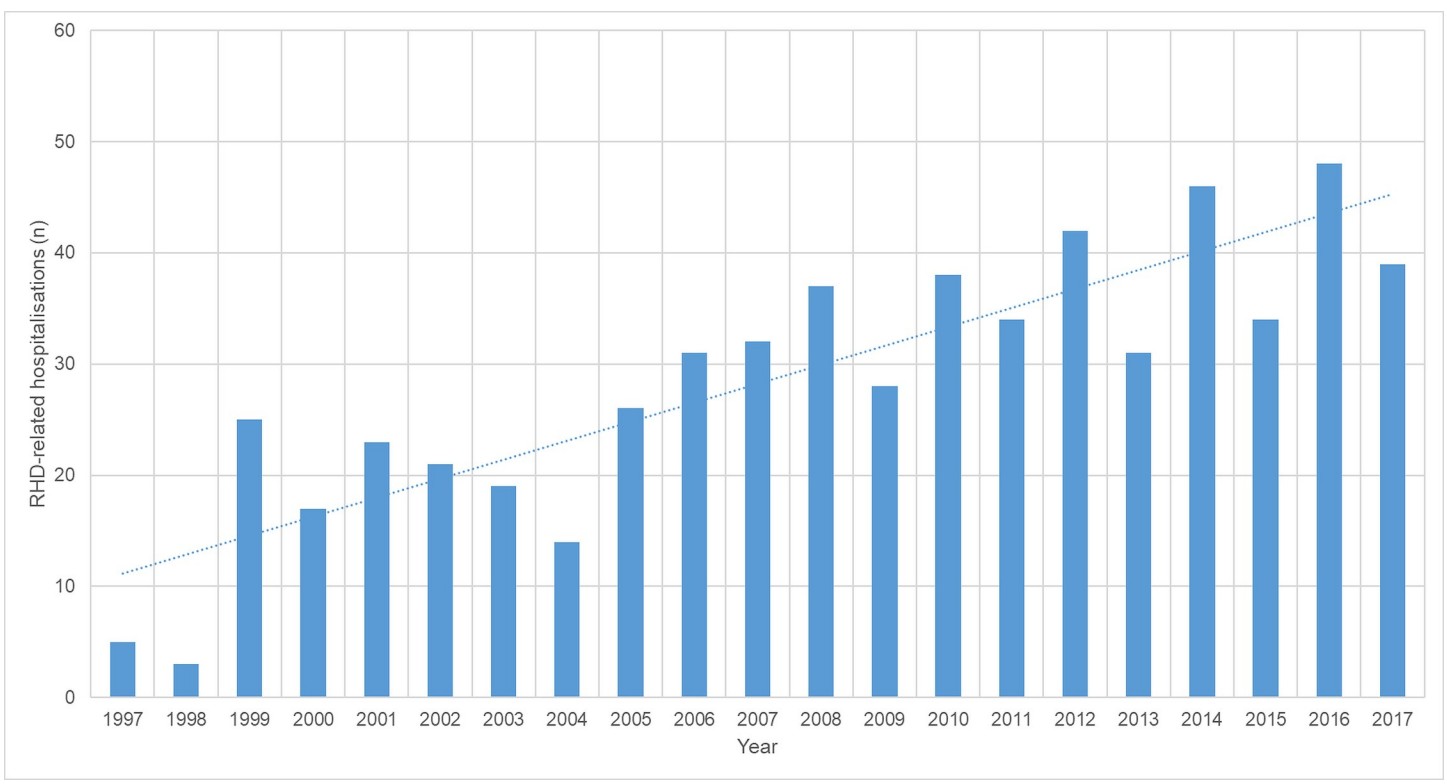

**Fig 6. RHD-related hospitalisations by year.**

**Table 4. Description of valvular interventions.**

| Procedures | Mitral valve | Aortic valve | Tricuspid valve | Pulmonary valve |
|---|---|---|---|---|
| Metallic valve replacement | 61 | 40 | 0 | 0 |
| Bioprosthetic valve replacement | 26 | 21 | 3 | 0 |
| Valvuloplasty | 39 | 6 | 17 | 1 |
| Total procedures | 126 | 67 | 21 | 1 |

Absolute numbers are presented.

status, SEIFA score and residence in a remote/rural location, only disease severity (p<0.001) and SEIFA score (p<0.001) remained significantly associated with surgery.

Indigenous patients had surgery at a younger age than non-Indigenous patients (median (IQR) age at the time of first surgical intervention: 33 (19–47) versus 54 (35–64), (p = 0.001). There were 51 women of child-bearing age (13–50) who had surgery during the study period, 49 (96%) of whom were Indigenous.

## Mortality

Non-Indigenous patients were more likely to die than Indigenous patients (9/64 (14%) versus 30/622 (5%), p = 0.002), but Indigenous patients died at a younger age (median (IQR): 52 (35–67) versus 73 (62–77) p = 0.01). In multivariate analysis that considered age, disease severity, Indigenous status, SEIFA score and residence in a remote/rural location, only age (OR (95% CI): 1.06 (1.04–1.08), p<0.001) was significantly associated with death.

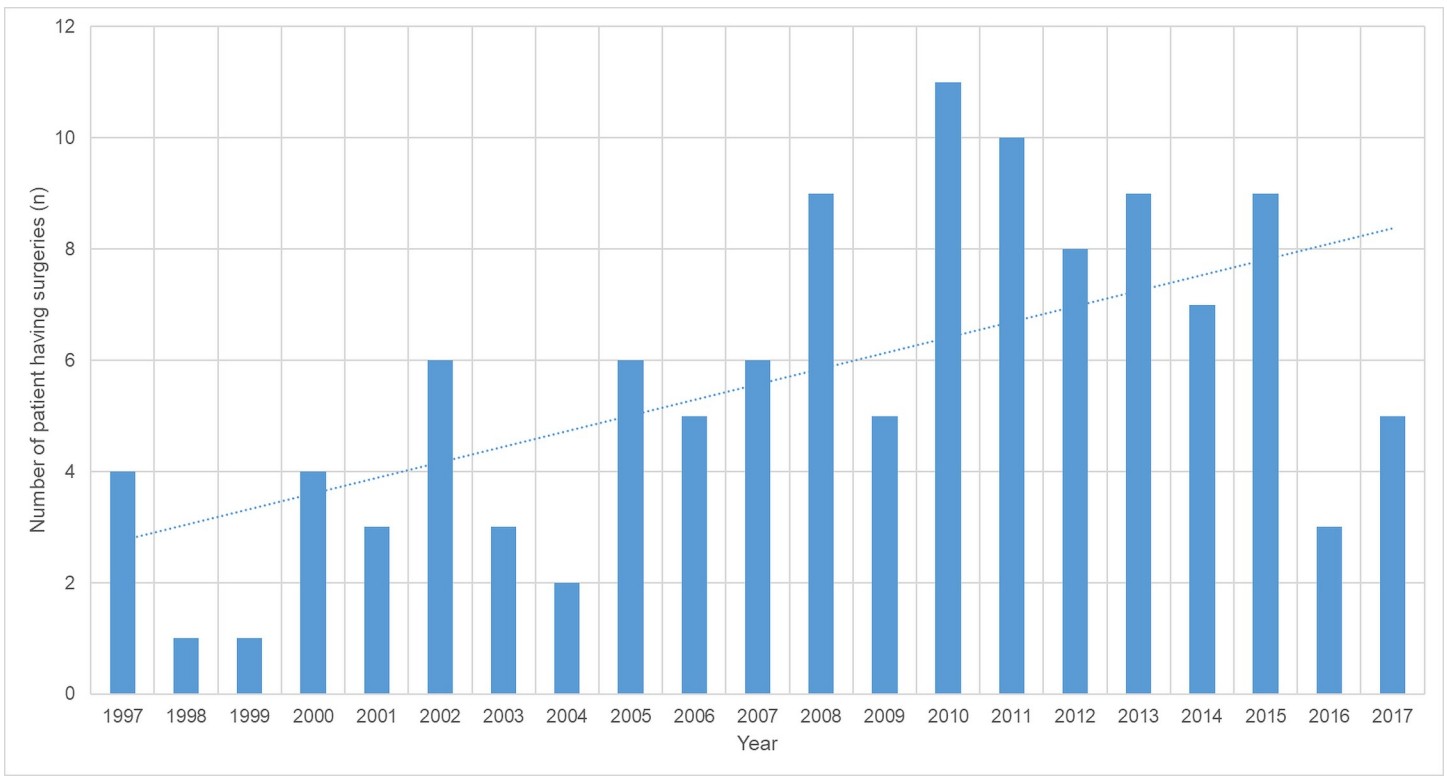

**Fig 7. Number of individual patients undergoing surgical procedures over time.**

**Table 5. Characteristics of the Indigenous Australians in the cohort stratified by whether the patient identified as an Aboriginal Australian or a Torres Strait Islander Australian.**

| Variable | Aboriginal Australian n = 347 | Torres Strait Islander Australian n = 205 | p |
|---|---|---|---|
| Age at the end of the study | 36 (23–51) | 31 (23–45) | 0.054 |
| Age at RHD diagnosis | 30 (19–46) | 24 (16–39) | 0.005 |
| Female | 233 (67.2%) | 141 (68.8%) | 0.69 |
| Rural/remote residence | 165 (47.6%) | 124 (60.5%) | 0.003 |
| Severe disease | 90 (25.9%) | 41 (20%) | 0.11 |
| Hospitalisations/person | 6 (3–12) | 4 (2–8) | 0.0001 |
| RHD-related hospitalisations | 0 (0–1) | 0 (0–1) | 0.36 |
| Valve surgery or valvuloplasty | 56 (16.1%) | 26 (12.7%) | 0.27 |
| Died | 17 (4.9%) | 13 (6.3%) | 0.47 |
| Age at death | 57 (37–68) | 51 (29–69) | 0.59 |
| Adherence | 39% (23–56) | 42% (25–61) | 0.24 |
| Good adherence | 35/149 (23.5%) | 24/90 (26.7%) | 0.58 |
| Number of echocardiograms | 6 (3–9) | 5 (3–7) | 0.0007 |
| RHD progression on serial echocardiograms | 87/328 (26.5%) | 24/181 (13.3%) | 0.001 |
| Specialist review | 329 (94.8%) | 188 (91.7%) | 0.15 |
| Dental revie | 105 (30.3%) | 60 (29.3%) | 0.81 |
| SEIFA score of residential address | 870 (713–960) | 836 (836–967) | 0.17 |

All numbers are median (interquartile range) or the absolute number (%)

Of the 39 deaths, 15 (38%) were linked directly to RHD: 9 from heart failure, 2 from infective endocarditis, 2 from a stroke, 1 from bleeding complications of warfarin therapy and 1 from complications of valve surgery. Of the 15 RHD-related deaths, 9 (60%) occurred in Indigenous patients. The median (IQR) age of Indigenous patients dying from RHD was 29 (12–58) years compared with 77 (64–78) among non-Indigenous patients dying from RHD (p = 0.007). Of the remaining 24 deaths, 17 were not RHD-related, while in 7 the patient's medical record could not be accessed. Of the 17 non-RHD related deaths, 14 (82%) occurred in Indigenous patients. Of these 14 deaths, 5 were related to cancer, 4 were from sepsis, 2 were from end stage renal disease, 1 was from ischaemic heart disease, 1 was from trauma and 1 was from dementia. The median (IQR) age of Indigenous patients dying from non-RHD causes was 59 (46–69) years compared with 62 (61–68) among non-Indigenous patients (p = 0.57).

## Comparison of Aboriginal and Torres Strait Islander Australians

Torres Strait islander Australians were more likely to live in a remote location than Aboriginal Australians but were less likely to have disease progression on serial echocardiograms and less likely to require hospitalisation. However, there were few other differences between the two populations (Table 5).

## Discussion

The incidence of new RHD diagnoses is rising in FNQ and Indigenous Australians living in the region's rural and remote locations continue to bear the greatest burden of disease. The number of RHD-related hospitalisations is also increasing, as is the number of patients requiring valve surgery. Whilst this is likely to be partly explained by greater disease recognition and enhanced service delivery, the strong association between socioeconomic disadvantage and RHD prevalence in the study and the absence of any diminution in new ARF diagnoses,

suggests that a greater focus on primordial and primary prevention is necessary if we are to reduce the significant impact of RHD in the region.

Although Australia has a well-resourced, universal healthcare system, the prevalence of RHD varies considerably between–and even within–regions in Australia [6]. This is a result of the complex interplay of local socioeconomic and sociocultural factors, differences in health seeking behaviour and heterogeneous trans-jurisdictional strategies that influence the delivery of care to Australia's geographically dispersed population [7]. All too predictably, Aboriginal and Torres Strait Islander Australians were over-represented in this cohort. Despite Indigenous Australians comprising 17% of the local population, they accounted for over 90% of the people living with RHD in the region. The prevalence of 27/1000 in Western Cape York, a region home to several Indigenous communities, is comparable to that seen in some of the poorest countries in the world [12,22–27]. The median age of Australian born non-Indigenous individuals living with RHD was 60, highlighting the fact that the disease was successfully countered with improvements in socioeconomic conditions and interventions that occurred in the 20th Century [7]. The median age of Indigenous Australians living with RHD was, by comparison, 33.

The significant variation in RHD prevalence across the region was linked strongly to socio-economic disadvantage. Furthermore, patients living in socioeconomically disadvantaged areas were not only more likely to have RHD, but they were also less likely to receive surgery, the only intervention that can definitively address established, advanced disease [7]. This para-doxical situation is not limited to RHD. The local Indigenous population, particularly those living in remote, socioeconomically disadvantaged locations bear a disproportionate burden of both infectious and non-communicable diseases, but also has less access to sophisticated healthcare [14,16,28–33]. And this captures the essence of the challenges in addressing the life expectancy gap between Indigenous and non-Indigenous Australians. Between one third and one half of this "gap" is explained by differences in the social determinants of health, while up to 43% of the difference in life expectancy can be explained by poorer access to health services [8]. Successive state and federal governments have attempted to address this complex issue, however it is generally agreed there has been very limited success [34].

In the case of RHD, there is uncertainty about where it is best to start. Although the rela-tionship between socioeconomic disadvantage and RHD is clearly established, it is less certain which social and environmental factors are the most significant in pathogenesis, nor is it clear how best to address these factors cost-effectively [3,22]. Crowding (household or other set-tings), income, dwelling characteristics, education level, employment, and nutrition have all been examined, with crowding consistently identified as not only one of the most important, but also one of the most amenable to improvement [3]. Crowded households have 1.7–2.8 times the risk of GAS infections, ARF, and RHD than uncrowded households, and also have higher rates of respiratory, gastrointestinal, eye and ear diseases [3,35]. Crowding also facili-tates the potentiation of scabies infection which is a major driving force of streptococcal pyo-derma in many Indigenous Australian communities [36,37].

Over 30% of Indigenous Australians in remote Queensland locations are living in over-crowded households [38]. In 2001, the Queensland Government implemented the Aboriginal and Torres Strait Islander Environmental Health Plan, a suite of environmental health inter-ventions that aimed to improve the health and wellbeing of Indigenous communities. Greater construction and improved maintenance of housing is one of the key initiatives of this pro-gram which also addresses sanitation, vector control, food hygiene and animal management [39]. There are some data to suggest that these environmental interventions are beginning to have an impact with the rates of leprosy and strongyloidiasis declining sharply in the region during this period [14,15]. However, these infections are arguably far easier to address than

GAS which are part of the normal human skin flora. It will be necessary to follow ARF notifications closely and to consider active surveillance which could be used to expedite more targeted action [40].

It is important to note that the rising incidence of RHD and continuing identification of ARF in the region is likely to be partly due to increased recognition from an expanded RHD program [41]. ARF was declared a notifiable disease in the state of Queensland in 1999, a time when education of local healthcare workers and enhanced surveillance for ARF was also commenced [42]. Echocardiography services have also increased: in 2011 a paediatric cardiologist began outreach services to the region, and since 2014 there has been an augmented adult specialist physician outreach service. New RHD diagnoses in the rural and remote communities served by these programs increased significantly, particularly among young people, after this expansion (Fig 3). The recent boosting of cardiology outreach services are also expected to improve medical and surgical management [43]. This strengthening of the medical effector arm of the RHD program is welcome as the cohort's median age is 34 and it would be expected that a significant proportion will develop complicated disease in the decades ahead [44].

Improving patients' access to medical care will reduce the morbidity and mortality attributable to RHD [45,46]. Indeed, there is some early evidence that the RHD program is reaching its most vulnerable patients. Patients in rural and remote locations are more likely to have received specialist review than patients living in urban areas, as are those in socioeconomically disadvantaged regions when compared to locations of greater affluence. Indeed, the expanded adult and paediatric specialist outreach programs and the resulting lower threshold for echocardiography is likely to explain why the incidence of RHD appears to be rising significantly despite the incidence of ARF remaining stable. It is also likely to explain why mild-moderate RHD was identified more commonly in remote, socioeconomically disadvantaged communities. This earlier recognition of RHD allows prompt prescription of secondary prophylaxis to reduce the risk of disease evolution [47,48]. Indeed, adherence to secondary prophylaxis in rural and remote communities and socioeconomically disadvantaged areas was also higher in this cohort. These data suggest that the local RHD program's efforts to educate health workers and provide care to patients in regions with the highest disease burden have been at least somewhat successful.

However, the fact that young people are still dying from RHD and that cases of ARF are still being diagnosed in 21st Century Australia, emphasises that the current approach needs to evolve. Even with an expanded RHD program, only 24.9% of the patients in the cohort prescribed secondary prophylaxis had good adherence at the end of the study period, meanwhile less than 30% had received a documented dental review, increasing their risk of infective complications. Practical guidelines have been developed for the recognition and management of ARF and RHD in Australia, however the challenge is now to translate these guidelines into practice that result in improvements in clinically meaningful endpoints [7,49]. To achieve this end it is essential to have adequately trained and supported Aboriginal and Torres Strait Islander workforce as they are more likely to have an insight into the personal, community, organisational and environmental factors that may influence engagement with care, particularly early presentation with GAS infection or long-term adherence to secondary prophylaxis [7,50]. The high rates of RHD among young women also highlight the need for a skilled female health workforce, including Aboriginal and Torres Strait Islander Health Workers, nurses, midwives and doctors [18].

This study was performed in the only part of Australia which has the homelands of both Aboriginal and Torres Strait Islander Australians, peoples that are frequently conflated but who have very different cultural histories. It is one of the largest to examine the burden of RHD in Torres Strait Islander Australians and to compare their longitudinal care and clinical course with that of Aboriginal Australians. It is essential to recognise that there is great

diversity within these two broadly described populations, but it is notable that their burden of RHD was similar. Despite these similarities, prospective examination of the barriers and enablers of RHD prevention and treatment in the ethnologically distinct Torres Strait Islander population is also needed if we are to address RHD in this population adequately.

This retrospective study has several limitations and almost certainly underestimates the local RHD burden. A determination of the true RHD prevalence would require systematic community screening [51], but this has never been performed in adults in the region. Although registry data were used to identify patients for this study, RHD was only declared a notifiable disease in September 2018, meaning that cases are likely to have been missed. Even though ARF was declared a notifiable disease in 1999—early in the study period—the addition of a patient to the register requires the treating clinician to contact the local public health unit; this is an imperfect process which will again underestimate disease incidence. Although this study aimed to examine the impact of socioeconomic status on disease burden, severity and access to care, registry data have a clinical focus and do not capture the environmental, economic, social or behavioural data which would influence these endpoints. Although the SEIFA score is determined by the Australian Bureau of Statistics using census data, it is calculated for entire regions rather than for individual residents. This may explain why there was greater correlation between the SEIFA score and the prevalence of RHD in different communities than the clinical endpoints in individual patients. Some data that were collected in private health settings–particularly dental review–were not accessible. This will tend to underestimate the amount of dental care patients were able to receive, although as private dental services are unavailable in most of the region, it is almost certainly still the case that local dental services also need to be improved, particularly for those with RHD. Finally, there are inherent limitations in using ICD coding to define the underlying cause of a patient's hospitalisation [52,53].

However, notwithstanding these deficiencies, the study uses almost over two decades of longitudinal data collected in both community and hospital settings to provide an overview of the scale of the problem of RHD in this unique part of Australia. It identifies many of the challenges that clinicians, public health physicians and governments will have in addressing a disease that persists so stubbornly in 21st century Australia.

## Conclusions

The incidence of RHD and RHD-related hospitalisations and surgery continues to rise in FNQ. Although this is likely to be partly explained by increased disease recognition and enhanced service delivery, the incidence of the disease remains unacceptably high and is almost entirely borne by the socioeconomically disadvantaged, Indigenous population. The current model of RHD care must evolve—in partnership with Indigenous communities—to have a greater focus on primordial prevention, and to improve access to care. This is not only likely to reduce the burden of RHD, but also the burden of many other diseases that are disproportionately borne by Indigenous Australians.

## Supporting information

**S1 STROBE Checklist. Strobe checklist**
(DOCX)

## Acknowledgments

The authors would like to acknowledge the work of all the healthcare workers involved in the care of patients living with RHD in Far North Queensland during the study period.

## Author Contributions

**Conceptualization:** Katherine Kang, Ken W.T. Chau, Josh Hanson.

**Data curation:** Katherine Kang, Ken W.T. Chau.

**Formal analysis:** Katherine Kang, Ken W.T. Chau, Josh Hanson.

**Investigation:** Katherine Kang, Ken W.T. Chau, Erin Howell, Josh Hanson.

**Methodology:** Katherine Kang, Ken W.T. Chau, Josh Hanson.

**Resources:** Erin Howell, Mellise Anderson.

**Supervision:** Simon Smith, Greg Starmer, Josh Hanson.

**Visualization:** Josh Hanson.

**Writing – original draft:** Katherine Kang, Josh Hanson.

**Writing – review & editing:** Ken W.T. Chau, Erin Howell, Mellise Anderson, Simon Smith, Tania J. Davis, Greg Starmer, Josh Hanson.

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
