## [Decision Letter · Decision Letter 0]

21 Oct 2020

Dear Dr Hanson,

Thank you very much for submitting your manuscript "The temporospatial epidemiology of rheumatic heart disease in Far North Queensland, tropical Australia 1997-2017; impact of socioeconomic status on disease burden, severity, and access to care" for consideration at PLOS Neglected Tropical Diseases. As with all papers reviewed by the journal, your manuscript was reviewed by members of the editorial board and by several independent reviewers. The reviewers appreciated the attention to an important topic. Based on the reviews, we are likely to accept this manuscript for publication, providing that you modify the manuscript according to the review recommendations. 

Sincerely,

Katja Fischer

Guest Editor

Joseph Vinetz

Deputy Editor

Please address all reviewer comments in your revised manuscript: 

Reviewer's Responses to Questions

**Key Review Criteria Required for Acceptance?**

**Methods**

-Are the objectives of the study clearly articulated with a clear testable hypothesis stated?

-Is the study design appropriate to address the stated objectives?

-Is the population clearly described and appropriate for the hypothesis being tested?

-Is the sample size sufficient to ensure adequate power to address the hypothesis being tested?

-Were correct statistical analysis used to support conclusions?

-Are there concerns about ethical or regulatory requirements being met?

Reviewer #1: Attached.

Reviewer #2: no comments

Reviewer #3: Methods seem appropriate to the objectives of the study. It is anticipated that the Australian Indigenous population residing in the remote areas will have other health conditions additional to RHD. It would be of interest to consider other relevant diseases/conditions from patients health records.

Reviewer #4: (No Response)

**Results**

-Does the analysis presented match the analysis plan?

-Are the results clearly and completely presented?

-Are the figures (Tables, Images) of sufficient quality for clarity?

Reviewer #1: attached

Reviewer #2: There is a decline in surgeries 2016-2017 that is quite significant compared to the upward moving trend from 1997 - 2015. Author should address why this may be the case - or at least discuss. Statement is true that number of surgeries increased WITHIN the study period (97-15) but not OVER the entire study period (97-17).

Reviewer #3: Should the numbers of ARF diagnosis (Fig 1) be presented in the same scale as RHD cases (Fig 2)? It is difficult to compare the two values otherwise to see the correlation.

Under Mortality: It is stated that although more numbers of non-indigenous patients died (14%) during the study period than indigenous patients (5%), indigenous patients died at a younger age. Only 38% of death was directly linked to RHD, 43% was not RHD-related, and 18% had no medical record. 

Could the authors clarify how many of RHD related death (e.g. out of 38%) were indigenous patients? It is possible that the indigenous patients died at a younger age due to health conditions/factors other than RHD while non-indigenous patients died from RHD? If that is the case, abstract should be amended to keep the indigenous age factor with death due to RHD in context.

Reviewer #4: (No Response)

**Conclusions**

-Are the conclusions supported by the data presented?

-Are the limitations of analysis clearly described?

-Do the authors discuss how these data can be helpful to advance our understanding of the topic under study?

-Is public health relevance addressed?

Reviewer #1: attached

Reviewer #2: no comments

Reviewer #3: The authors discussed in details about the impact of overcrowding in the remote indigenous communities. Risks for infection GAS, ARF, RHD and other respiratory, gastrointestinal, eye and ear diseases. Scabies is a highly prevalent skin disease among remote indigenous communities in North Queensland. It would be appropriate to discuss scabies as possible risk factor for GAS infections and secondary complications such as ARF and RHD.

Line 409: Please include explanation for the significance of dental review. Is dental review representative of access to health care in these communities?

Line 442: limitation in using ICD coding to define patient’s hospitalisation was discussed. Please provide a brief explanation of what ICD coding is and why this is a limitation.

Reviewer #4: (No Response)

**Editorial and Data Presentation Modifications?**

Reviewer #1: (No Response)

Reviewer #2: Two minor grammar errors line 

- line 3 of the Introduction:

Improvements in the standard of living and the development of more effective treatment of group A streptococcal (GAS) infections in (remove in) during the 20th century

- line 1 of Comparison of Aboriginal and Torres Strait Islander Australians - change islander to Islander (capital I).

Reviewer #3: Accept.

Reviewer #4: (No Response)

**Summary and General Comments**

Reviewer #1: (No Response)

Reviewer #2: Excellent paper, well written and concise. Will add new knowledge to the subject area and provide advocacy for change.

Would recommend formally acknowledging the contribution of the RHD Register and Control Program in an Acknowledgements section rather than in the methods section. Alternatively, if the significant contribution was from individual(s) could consider offering an authorship. This is becoming an appropriate/respectful way of recognising the contribution of community workers/organisations.

In the introduction, elimination of malaria/filariasis/strongyloides/leprosy is mentioned. May want to consider discussing that in addition to RHD, scabies is still endemic in many regions where RHD occurs and efforts to address scabies are needed. Especially as scabies is considered to contribute to RHD development.

Reviewer #3: Manuscript is well written. This is a retrospective analysis of prevalence or Rheumatic Heart Disease (RHD) in far North Queensland using RHD registry (1997-2017). Although high prevalence of RHD in the remote indigenous communities are well-known, it is confronting to see the dispropionately high rates of RHD in younger age group. 

It would be interesting for the authors to speculate why RHD is highly prevalent in younger age group for indigenous community compared with non-indigenous population. Could the authors also explain why despite high prevalence of RHD, almost half of RHD indigenous cases are mild to moderate (48%, 27% respective) and only 24% severe.

Reviewer #4: 1. Figure 1, 2 and 3 - To make the graph more informative, I would like to suggest to have new RHD admissions of the each year to be indicated in a different colour in the same bar of total RHD cases.

2. It would be good to discuss the potential reasons to have low RF specifically in the years with high RHD. In addition, if the increased health services have contributed to high RHD cases in the recent year, how authors would describe the low/steady case numbers of RF?

PLOS authors have the option to publish the peer review history of their article (what does this mean?). If published, this will include your full peer review and any attached files.

Reviewer #1: No

Reviewer #2: No

Reviewer #3: No

Reviewer #4: No
---

## [Editor Report · Decision Letter 1]

16 Nov 2020

Dear Dr Hanson,

We are pleased to inform you that your manuscript 'The temporospatial epidemiology of rheumatic heart disease in Far North Queensland, tropical Australia 1997-2017; impact of socioeconomic status on disease burden, severity, and access to care' has been provisionally accepted for publication in PLOS Neglected Tropical Diseases.

Best regards,

Katja Fischer

Associate Editor

Joseph Vinetz

Deputy Editor

---

## [Editor Report · Acceptance letter]

8 Jan 2021

Dear Dr Hanson,

We are delighted to inform you that your manuscript, "The temporospatial epidemiology of rheumatic heart disease in Far North Queensland, tropical Australia 1997-2017; impact of socioeconomic status on disease burden, severity, and access to care," has been formally accepted for publication in PLOS Neglected Tropical Diseases.

Best regards,

Shaden Kamhawi

co-Editor-in-Chief

Paul Brindley

co-Editor-in-Chief
